# Use of Composite Multivariate Multiscale Permutation Fuzzy Entropy to Diagnose the Faults of Rolling Bearing

**DOI:** 10.3390/e25071049

**Published:** 2023-07-12

**Authors:** Qiang Yuan, Mingchen Lv, Ruiping Zhou, Hong Liu, Chongkun Liang, Lijiao Cheng

**Affiliations:** 1School of Naval Architecture, Ocean and Energy Power Engineering, Wuhan University of Technology, Wuhan 430070, China; 2School of Naval Architecture and Maritime, Zhejiang Ocean University, Zhoushan 316022, China

**Keywords:** fault diagnostic method, multivariate multiscale permutation fuzzy entropy, composite multivariate multiscale permutation fuzzy entropy, rolling bearing

## Abstract

The study focuses on the fault signals of rolling bearings, which are characterized by nonlinearity, periodic impact, and low signal-to-noise ratio. The advantages of entropy calculation in analyzing time series data were combined with the high calculation accuracy of Multiscale Fuzzy Entropy (MFE) and the strong noise resistance of Multiscale Permutation Entropy (MPE), a multivariate coarse-grained form was introduced, and the coarse-grained process was improved. The Composite Multivariate Multiscale Permutation Fuzzy Entropy (CMvMPFE) method was proposed to solve the problems of low accuracy, large entropy perturbation, and information loss in the calculation process of fault feature parameters. This method extracts the fault characteristics of rolling bearings more comprehensively and accurately. The CMvMPFE method was used to calculate the entropy value of the rolling bearing experimental fault data, and Support Vector Machine (SVM) was used for fault diagnosis analysis. By comparing with MPFE, the Composite Multiscale Permutation Fuzzy Entropy (CMPFE) and the Multivariate Multiscale Permutation Fuzzy Entropy (MvMPFE) methods, the results of the calculations show that the CMvMPFE method can extract rolling bearing fault characteristics more comprehensively and accurately, and it also has good robustness.

## 1. Introduction

Rolling bearings are one of the most important components of mechanical equipment, and their safety and reliability are crucial for the operation of machines and industrial production. Due to long-term use and complex environmental factors, rolling bearing failures are inevitable and can lead to shortened remaining useful life or damage and casualties of property [1,2,3]. As a result, demand for bearing failure diagnosis is also increasing. The timely diagnosis of defects and the choice of repairs or replacements are beneficial in theory and practice for the safety of production [4].

Currently, academics in different countries are conducting extensive research into rolling bearing problems. Among them, the extraction of the characteristics of fault parameters is a key link in the process of using machine learning methods for fault diagnosis research. Rolling bearing failure signals have features such as nonlinearity, periodic transience, and low signal-to-noise ratio. It is difficult to apply common methods such as time domain analysis, frequency domain analysis, and time-frequency domain analysis to these features. Therefore, it is proposed to utilize the statistical analysis advantage of entropy in the field of rotating machinery fault diagnosis. Specifically, by calculating the entropy value of rolling bearing fault data, a more comprehensive and accurate assessment of the complexity of the fault data can be achieved. This approach effectively reflects the fault characteristics of the bearings and improves the efficiency of rotating machinery fault classification. Thus, it serves as a highly effective method for extracting features from rolling bearing fault data [5,6].

Zhuang et al. [7], Minhas et al. [8], and Mantas et al. [9] have been successfully applied to mechanical error diagnosis entropy algorithms such as sample entropy, fuzzy entropy, and permutation entropy in a series of ways [10]. However, the above-mentioned research only analyzed fault signals on a single scale, ignoring information implied by different scale factors. Rohit et al. [11] combined multiscale permutation entropy and adaptive neuro-fuzzy classifier to extract MPE features and demonstrated the potential of the proposed method in diagnosing early bearing faults. Multiscale overcomes the defect that entropy calculations under a single scale condition are not sufficient to analyze time series comprehensively. However, there are some limitations in processing only a single index of entropy at multiple scales, ignoring information contained in other coarse-grained sequences under the same scale factor. Traditional multiscale algorithms are limited by data length. As the scale increases, entropy errors increase gradually, and the probability of unknown entropy increases. Mohamed et al. [12] combined the fuzzy entropy of empirical modal decomposition, principal component analysis, and self-organizing graph neural network for the fault diagnosis of bearings, and the results not only correctly assessed the degradation of rolling bearings but also identified highly sensitive defects for different types of bearing failures. Jin et al. [13] proposed an improved segmented multivariable multiscale fuzzy entropy as a flawed feature with a rate of up to 99% flaw detection. Coarse-grained time selects composite coarse-grained time to solve the problem of losing other coarse-grained sequences under the same scale factor. Furthermore, in complex composite coarse grains at different scales, only coarse grains based on mean cause a loss of useful information, resulting in a calculated entropy value that cannot fully and accurately characterize the complexity of the time series [14]. As a result, the poly coarse-grained form was proposed to solve the above problems [15,16,17].

This paper presents a novel method for calculating permutation fuzzy entropy by combining the high calculation accuracy of fuzzy entropy with the strong noise resistance of permutation entropy. The proposed method incorporates the concept of sorting symbolization into the calculation process of fuzzy entropy. The permutation fuzzy entropy calculation method is enhanced through composite multiscale processing to address the issue of unstable entropy calculation at large scale factors. To ensure the comprehensive and accurate extraction of information from fault signal samples, the composite multiscale permutation fuzzy entropy method is utilized. The article introduces the concept of a multivariate coarse-grained method as a solution to the problem of loss of fault feature information during entropy calculation. The three coarse-grained forms based on mean, root, mean square, and variance are presented, and a Composite Multivariate Multiscale Permutation Fuzzy Entropy (CMvMPFE) method is established. The article then applies this model to calculate the entropy value of the extraction of fault features for four bearing states’ data on the public rolling bearing test bench of Case Western Reserve University in the United States, and the results of the experimental data analysis indicate that the CMvMPFE model proposed in this article is capable of efficiently extracting four bearing state feature parameters and accurately distinguishing four states. The calculations demonstrate that this model has a high level of accuracy in entropy calculation and is also highly resistant to noise. Furthermore, the model takes into account the stability, continuity, and completeness of the entropy calculation. Overall, the research demonstrates the effectiveness of CMvMPFE in characterizing bearing states, making it a valuable tool for studying rolling bearing fault feature extraction and identification. The study covers the following main topics:(1)In this paper, a new fault feature extraction entropy calculation method called CMvMPFE is proposed for signals with non-linear characteristics, periodic impact, and low signal-to-noise ratio. The method is based on fuzzy entropy and permutation entropy and introduces the concepts of composite multiscale and multivariate coarse-grained.(2)The MPFE, CMPFE, MvMPFE, and the proposed CMvMPFE methods are used to calculate entropy values and extract features from four states of bearing test data. The extracted feature parameters are divided into test data and training data. The SVM model is trained using the training data, and the trained SVM is used to classify and identify the bearing states of the experimental data. The accuracy of fault identification is compared and analyzed.(3)This study utilizes the CMvMPFE proposed in this article to perform fault feature extraction entropy calculation and analysis of different distribution locations of the same fault type and different fault depths of the same fault location.

The rest of this paper is organized as follows. Section 2 introduces the composite multivariate multiscale permutation fuzzy entropy method and basic fuzzy entropy and permutation entropy methods and introduces the CMvMPFE-SVM fault diagnosis method. Section 3 introduces the relevant feature extraction results and the CMvMPFE-SVM diagnosis results based on the CWRU dataset. Section 4 introduces the robustness analysis results of the CMvMPFE (rms). Finally, the conclusion is drawn in Section 5.

## 2. Composite Multivariate Multiscale Permutation Fuzzy Entropy

### 2.1. Fuzzy Entropy and Permutation Entropy

#### 2.1.1. Fuzzy Entropy

Fuzzy entropy (FE) is an improvement upon sample entropy, where the unit step function in sample entropy is replaced by the exponential function $e^{-\left(d/r\right)}{}^{n}$, known as the fuzzy function. This replacement ensures the continuity of entropy values and maximizes the self-similarity values of the vectors. It is used to measure the similarity between two vectors. The definition is as follows:

(1) For a time series $u\left(i\right)=\left\{u\left(1\right),u\left(2\right),\ldots ,u\left(N\right)\right\}$ of length N, the first step is to determine the vector dimension m. Next, we reconstruct the time series u to obtain a new sequence:(1)$$X_{i}^{m}=u\left(i\right),u\left(i+1\right),\ldots \;,u\left(i+m-1\right)-u_{0}\left(i\right),\;i=1,2,\ldots ,N-m+1.$$

The continuous sequence of m elements starting from the *i*-th point, where m represents the embedding dimension, is subtracted by the mean value $u_{0}\left(i\right)$, where
(2)$$u_{0}\left(i\right)=\frac{1}{m}\sum_{j=0}^{m-1}\;u\left(i+j\right).$$

(2) The maximum distance $d_{ij}^{m}$ between two reconstruction vectors $X_{i}^{m}$ and $X_{j}^{m}$ is defined as $d_{ij}^{m}=d\left[X_{i}^{m},X_{j}^{m}\right]$.
(3)$$\begin{array}{c} d_{ij}^{m}=d\left[X_{i}^{m},X_{j}^{m}\right]=\underset{k\in \left(0,m-1\right)}{max}\left|\left(u\left(i+k\right)-u_{0}\left(i\right)\right)-\left(u\left(j+k\right)-u_{0}\left(j\right)\right)\right|\\ i,j=1,2,\ldots ,N-m,i\neq j \end{array}$$

(3) Fuzzy membership function is defined as:(4)$$\;\;\;\;\;A_{ij}^{m}=e^{\left[-{\left(\frac{d_{ij}^{m}}{r}\right)}^{n}\right]}.$$

The fuzzy membership function is an exponential function, with n representing its boundary gradient and r representing the similarity tolerance, typically taken within the range of 0.1 to 0.25 times the standard deviation (SD) of the original data.

(4) We define the function as follows:(5)$$\;\;C_{i}^{m}n,r=\frac{\sum_{j=1,j\neq i}^{N-m+1}\;A_{ij}^{m}}{N-m};$$

Therefore, we obtain
(6)$$\phi^{m}\left(n,r,N\right)=\frac{\sum_{i=1}^{N-m+1}\;C_{i}^{m}\left(n,r\right)}{N-m+1}.$$

(5) Similarly, after increasing the vector dimension to m + 1, we can repeat the above steps to obtain $\phi^{m+1}\left(n,r,N\right)$:(7)$$\;\phi^{m+1}\left(n,r,N\right)=\frac{\sum_{i=1}^{N-m+1}\;C_{i}^{m+1}\left(n,r\right)}{N-m+1}.$$

(6) Fuzzy entropy is defined as follows:(8)$$FE(m,n,r,N)=ln\phi^{m}\left(n,r,N\right)-ln\phi^{m+1}\left(n,r,N\right)$$

#### 2.1.2. Permutation Entropy

Permutation entropy (PE), similar to FE, was initially designed to quantify the complexity of a time series. However, PE does not consider the specific numerical values of the time series data. Instead, it introduces the notion of permutations based on comparisons of adjacent data points [18]. Its definition is as follows:

(1) For a time series of length N: $X\left(i\right),\;i=1,2,\ldots ,N$, perform phase space reconstruction to obtain the following time series matrix:(9)$$\left[\begin{array}{cccc} x\left(1\right) & x\left(1+t\right) & \ldots & x\left(1+\left(m-1\right)t\right)\\ x\left(2\right) & x\left(2+t\right) & \ldots & x\left(2+\left(m-1\right)t\right)\\ \ldots & \ldots & \ldots & \ldots \\ x\left(i\right) & x\left(i+t\right) & \ldots & x\left(i+\left(m-1\right)t\right)\\ \ldots & \ldots & \ldots & \ldots \\ x\left(K\right) & x\left(K+t\right) & \ldots & x\left(K+\left(m-1\right)t\right) \end{array}\right],i=1,2,\ldots ,K,$$
where t is the time delay, usually taken as 1; m is the permutation dimension, generally taken in the range of 3~7.  K=N −(m−1)t; and each row represents a reconstructed component, with a total of K reconstructed components.

(2) Sort the m data of each reconstructed component in ascending order; that is,
(10)$$\;\;x\left(i+\left(j_{1}-1\right)t\right)⩽x\left(i+\left(j_{2}-1\right)t\right)⩽\cdots ⩽x\left(i+\left(j_{m}-1\right)t\right).$$

If $x\left(i+\left(j_{i1}-1\right)t\right)=x\left(i+\left(j_{i2}-1\right)t\right)$, then sort according to the size of the value i. Thus, each reconstructed vector $X\left(i\right)$ is mapped to a new symbolic sequence $U\left(i\right)$.

(3) Calculate the probability P_i_ of each symbol sequence occurring, where ∑P=1, and I=1,2,…, k, k≤m!. There are m! different ways to arrange m different symbols.

(4) By analogy with the definition of information entropy, calculate permutation entropy as follows:(11)$$H_{p}\left(m\right)=-\sum_{i=1}^{k}\;P_{i}lnP_{i},$$
when $P_{i}=\frac{1}{m!}$, each symbol has an equal probability, and at this time, the permutation entropy $H\left(m\right)$ is at its maximum, which is $ln\left(m!\right)$.

(5) Normalization. For convenience, the entropy value range is adjusted to 0 to 1 and normalized:(12)$$H_{p}={\frac{H\left(m\right)}{ln\left(m!\right)}}_{\;}.$$

The variation of the permutation entropy ($H_{p}$) reflects and amplifies the local subtle changes in the time series. A higher value of $H_{p}$ indicates a higher level of randomness or complexity in the time series. Conversely, a lower value of $H_{p}$ suggests a more regular or predictable behavior in the time series.

### 2.2. Permutation Fuzzy Entropy

In this study, we proposed a Permutation Fuzzy Entropy (PFE) algorithm that combines the accuracy of fuzzy entropy with the simplicity and noise resistance of permutation entropy. The algorithm achieves this by introducing the symbolization idea of permutation entropy into fuzzy entropy. The first step of the algorithm involves performing a sorting symbolization on the original time series to generate a new sequence. To obtain the permutation fuzzy entropy of the original sequence, first, create a new sequence that reflects the immediate trend of the original time series through values between 1 and m!. This ensures that the trend of the new sequence is consistent with the trend or complexity of the original time series. Next, calculate its fuzzy entropy using the following specific calculation steps:

(1) According to (9), reconstruct the phase space of a time series of length N: $X\left(i\right),\;i=1,2,\ldots ,N$ to obtain a time series matrix.

(2) According to (10), sort the m data of each reconstructed component in ascending order. When j1<j2, there is $x\left(i-\left(j_{1}-1\right)t\right)\leq \;x\left(i-\left(j_{2}-1\right)t\right)$ [19,20]. In this way, after rearranging, a total of m! symbol sequences are obtained. The m! symbol sequences, respectively, corresponded to the values between 1 and m!, so that the original time series $X\left(i\right)$ is transformed into a new sequence $U\left(i\right),$ with each element taking values between 1 and m!:(13)$$U\left(i\right):1\leq i\leq N-\left(m-1\right)t.$$

(3) Calculate the fuzzy entropy of the new sequence $U\left(i\right)$ [21]; thus, the PFE of the original sequence can be obtained.

### 2.3. Composite Multivariate Multiscale Permutation Fuzzy Entropy

Composite Multiscale Permutation Fuzzy Entropy (CMPFE) is a method that combines coarse-grained sequences and PFE to extend single-scale PFE to multiscale MPFE, which reflects the complexity of multiscale time series. The method involves extending a single coarse-graining method to a multivariate coarse-graining method to reduce the loss of information in time series. The specific process is as follows.

For the new sequence $U\left(i\right)$ obtained after sorting and symbolizing as described above, it is subjected to coarse-graining to obtain the coarse-grained sequence $y_{j}\left(\tau \right)$:(14)$$\;y_{j}\left(\tau \right)=\frac{1}{\tau }\sum_{i=\left(j-1\right)\tau +1}^{j\tau }\;x_{i},1⩽j⩽\frac{N}{\tau },$$
where τ is the scale factor. When τ = 1, the coarse-grained sequence is the original sequence. N/τ represents the length of the sequence after coarse-graining at different time scales (τ > 1). The coarse-graining process when τ = 2 is shown in Figure 1.

As the scale factor increases, the length of the coarse-grained sequence decreases. This highlights the need for a sufficiently long time series to ensure accurate results in multiscale analysis. However, using MPFE may result in a loss of information from other coarse-grained forms at the same scale, leading to significant deviations in calculation results. Therefore, it is important to enhance the coarse-graining process of MPFE. When the scale factor is τ, the composite coarse-grained sequence $y_{k,j}^{\left(\tau \right)}$ is obtained:(15)$$y_{k,j}^{\left(\tau \right)}=\frac{1}{\tau_{\;}}\sum_{i=\left(j-1\right)\tau +k}^{j\tau +k-1}\;x_{i},1⩽j⩽\frac{N}{\tau },1⩽k⩽\tau ,$$
where k is the sliding sequence number at a certain time scale τ. For a single scale factor τ, only one coarse-grained sequence is generated, while the composite generates τ coarse-grained sequences by smoothing the moving order. Figure 2 illustrates the coarse-graining process when τ = 2.

To calculate the CMPFE, first, calculate the fuzzy entropy of τ coarse-grained sequences separately at the same scale factor; then, average the τ multiscale permutation fuzzy entropy values of multiscale permutation.
(16)$$CMPFE=\frac{1}{\tau }\overset{\tau }{\underset{k=1}{\sum }}\;PFE$$

The CMPFE method, based on mean coarse-graining, still has limitations in dealing with fault signals. To address this issue, a new multivariate coarse-graining method is proposed. This method utilizes three different forms of coarse-graining: mean, root mean square (RMS), and variance (VAR). The coarse-graining processes for each form are as follows.

Coarse graining based on MEAN:(17)$$y_{k,j}^{\left(\tau \right)}=\frac{1}{\tau }\sum_{i=\left(j-1\right)\tau +k}^{j\tau +k-1}\;x_{i}.$$

Coarse-graining based on RMS:(18)$$\;y_{k,j}^{\left(\tau \right)}=\sqrt{\frac{1}{\tau }\sum_{i=\left(j-1\right)\tau +k}^{j\tau +k-1}\;x_{i}^{2}}.$$

Coarse-graining based on VAR:(19)$$y_{k,j}^{\left(\tau \right)}=\frac{1}{\tau }\sum_{l=\left(j-1\right)\tau +k}^{j\tau +k-1}\;{\left(x_{i}-\bar{x}\right)}^{2},\bar{x}=\frac{1}{\tau }\sum_{i=\left(j-1\right)\tau +1}^{j\tau }\;x_{i}.$$

To ensure comprehensive analysis, this study analyzed the Multivariate Multiscale Permutation Fuzzy Entropy (MvMPFE) and compared it with the CMPFE and CMvMPFE feature entropy values. The four characteristic entropy values of MPFE, CMPFE, MvMPFE, and CMvMPFE are used to extract bearing fault characteristics and compare their effectiveness. The best representation is then selected as a fault characteristic for fault diagnosis.

### 2.4. Parameter Seslection

In the CMvMPFE algorithm, there are a total of 7 parameters involved: ① time delay τ; ② permutation dimension m; ③ embedding dimension M; ④ similarity tolerance r; ⑤ boundary gradient n of the fuzzy function; ⑥ scale factor τ; and ⑦ sample length N.

Based on previous studies [22,23], it is suggested to select a dimension of permutation m ranging from 3 to 7. After conducting experiments, we discovered that the optimal result was achieved when m = 7 and τ = 1. It is important to note that if r is too large, valuable information may be lost; while if it is too small, the desired effect may not be achieved. Typically, r is set to 0.1~0.25 SD (where SD represents the standard deviation of the original time series). In this paper, the similarity tolerance r = 0.15 SD is used. Increasing the embedding dimension M includes more information in the reconstruction process but also increases calculation length and amount. According to [24], M = 2 is used. The boundary gradient n is a weight in calculating the similarity between fuzzy entropy vectors. According to [25], it is set to n = 2. The scale factor can be set to a value greater than 10. To better observe the information on a larger scale, τ = 16 is taken here.

### 2.5. Diagnostic Process

The method for bearing fault diagnosis based on CMvMPFE (rms) and SVM involves the following steps:

(1) Signal acquisition, wherein bearing vibration signals of four states are collected, and three sets of data for each state are selected, with each set containing 2048 data points;

(2) Features extraction, wherein parameters are set, and MPFE, CMPFE, MvMPFE, and CMvMPFE are calculated for each set of data to construct a feature vector set;

(3) Fault diagnosis, wherein the obtained feature vector set is randomly divided into training and test sets. The training samples are used to train the SVM classifier model, and the test set is used to diagnose faults using the trained SVM model, resulting in the fault diagnosis outcome.

Figure 3 shows the fault diagnosis process.

### 2.6. CMvMPFE Procedure 

   CMvMPFE-SVM Procedure   load CWRU dataset   Data = data fragment with length 2048   tau = scale factor   for tau = 1 to 16   for i = 1 to tau   Select the ith to end data from data   Roughening of the selected data   Symbolize the coarse-grained data   Calculates the fuzzy entropy of the symbolized data   end for   Average the entropy values of tau   end for   SVM is used for fault diagnosis and the diagnostic accuracy of the extracted en tropy value features is calculated   Preprocess the data labels and divides the test and training sets   xlsread (feature vector set)   Train = training sample data   Test = Test sample dataBuild a support vector machine   Obtain the highest diagnostic rate of CMvMPFE and output the diagnostic results

## 3. Verification of Rolling Bearing Fault Feature Extraction Method Based on CMvMPFE

The proposed method, CMvMPFE, improves the accuracy of signal feature extraction compared to MPFE and CMPFE. The resulting entropy feature is more stable. The rolling bearing fault diagnosis method using CMvMPFE and SVM involves the following steps: 

Step 1: We selected vibration signal data for four states (inner ring failure (IR), outer ring failure (OR), rolling element failure (B), and normal (N)) based on the characteristics of the rolling bearing test bench data from Case Western Reserve University in the United States. 

Step 2: To analyze the influence of different entropy calculation methods on the extraction of fault signal feature samples, the entropy values of vibration signal samples from four rolling bearing states were calculated using four methods: MPFE, CMPFE, MvMPFE, and CMvMPFE. 

Step 3: The resulting entropy values were then plotted on characteristic curves, with the scale factor as the abscissa. Through this analysis, we compared and analyzed the effectiveness of the four entropy calculation methods.

### 3.1. Data Collection

To validate the proposed method, we utilized it for analyzing experimental data obtained from the Bearing Data Centre of Case Western Reserve University in the United States. The experimental test data utilized in this study pertained to rolling bearings. The test system is illustrated in Figure 4, with the 6205-2RSJEM SKF deep groove ball bearing being the model used at the drive end. The rolling bearing was subjected to electric spark machining to create fault points with a diameter of 0.5334 mm (21 mils). Vibration signals were collected at the drive end in four states: normal (N), inner ring (IR), outer ring (OR), and rolling element (B), at a speed of 1797 r/min and a sampling frequency of 12 KHz. A total of 80 sets of data were collected, with 20 sets for each state. It is important to note that the sample length cannot be too short as it can result in large errors in the multiscale process. To ensure accuracy in our analysis, we set the signal length of each state to 2048 data points and collected 20 sets of data for each state, resulting in a total of 80 sets. The resulting time-domain waveforms for normal (N), rolling element failure (B), inner ring failure (IR), and outer ring failure (OR) can be seen in Figure 5:

From Figure 5, the vibration signals of the three fault states exhibit obvious periodic shocks compared to the normal state. They have a certain degree of regularity, indicating that the fault signals are more self-similar. In addition, there is a distinguishable pattern of effects between different fault states. Therefore, entropy can be used to determine the condition of the bearing. During the process of acquiring signals, noise interference is a common problem, and the complexity of the signal is often high. Analyzing fault-related information at different time scales can be difficult, as there may not be clear differences between the four states on the time-domain waveform. To address this issue, three sets were randomly selected from each of the four states, and twelve sets of data were analyzed under different parameter conditions. The outer ring failure was selected in the 3 o’clock direction.

### 3.2. Signal Characteristic Analysis

Rolling bearings inevitably produce vibrations in both normal and faulty states. The fault signals of bearings manifest as periodic impact signals, where the occurrence of faults is reflected in changes in the frequency and amplitude of the original signal. The spectrograms of the four state signals are shown in the Figure 6. By analyzing the impulsiveness and cyclostationarity, we can gain insights into the presence of faults, anomalies, or disturbances in a system, aiding in diagnosis, prognosis, and decision-making for maintenance or corrective actions. Hence, based on the impulsiveness and gyroscopic stability of bearing vibration signals, Qing Ni et al. [2] proposed a new bearing prognosis scheme. These characteristics of bearing signals align well with the advantages of entropy calculation in statistical analysis. In addition, impulsiveness is also a valid bearing failure feature [26].


*Impulsiveness*


Impulsiveness refers to the characteristic of a signal or system that exhibits abrupt and sudden changes or impulses. It is associated with the occurrence of instantaneous and high-amplitude events within the signal. Impulsiveness can manifest as sharp spikes or spikes with short durations in the time-domain representation of the signal. It is often characterized by high peak amplitudes and rapid changes in signal values. Understanding the impulsiveness of signals provides valuable information about the dynamic behavior and integrity of systems, helping to ensure their reliability, safety, and performance.


*Cyclostationarity*


Cyclostationarity refers to the property of a signal where statistical properties exhibit periodic variations over time. The vibration signals of rolling bearings have periodic changes over time, and this change itself is also periodic, resulting in the generation of cyclic stability. In cyclostationary analysis, the focus is on studying the cyclostationary features of a signal, such as cyclic autocorrelation (see Figure 7) and cyclic power spectral density (see Figure 8). 

From Figure 7 and Figure 8, the periodicity of the bearing vibration signal during the cyclically stable process and the power under the main frequency components corresponding to each state can be reflected.

### 3.3. Feature Extraction

For the selected 12 sets of data, we calculated their MPFE, CMPFE, MvMPFE and CMvMPFE, respectively. The entropy curves obtained are shown in Figure 9.

The results from Figure 9a indicate that the MPFE curve is disorderly and can only differentiate between normal and faulty states. On the other hand, the entropy value curves for the three fault states are mixed and difficult to distinguish. Figure 9b shows that the CMPFE curve does not show significant improvement compared to the MPFE curve. However, the entropy value curve remains chaotic, making it challenging to effectively identify the bearing fault state. Consequently, the feature extraction for identifying faults is not effective. In comparison to the MPFE and CMPFE curves, the MvMPFE curve depicted in Figure 9c exhibits an improved ability to differentiate between different state signal samples. However, it is limited in its capacity to distinguish between four states, indicating the need for increased granularity and diversified analysis. The MvMPFE curve is able to distinguish between upper and lower positions to some extent and provides a general trend. However, the entropy value distribution of rolling bearing vibration signals in the same state remains relatively dispersed, which limits its effectiveness in extracting rolling bearing fault characteristics.

The comparison shows that the two CMvMPFE curves maintain the correct position distribution and trend of entropy values for different bearing state signals. However, the entropy values for the same state with the scale factor are more concentrated, indicating that CMvMPFE’s ability to represent faults is enhanced after compounding. It effectively distinguishes different bearing states and can be used as a rolling bearing fault feature. The CMvMPFE (var) curve lacks smoothness and information completeness when the scale factor is τ = 1. On the other hand, the CMvMPFE (rms) curve is optimal in terms of both curve smoothness and scale information completeness.

### 3.4. Fault Diagnosis

Based on the analysis, it is evident that CMvMPFE (rms) is an effective method for accurately extracting fault features of bearings. Therefore, in this study, the CMvMPFE (rms) of the signal sample was utilized as the input for the fault classifier for identification. A support vector machine was chosen as the classifier. The study collected 20 sets of vibration data for each type, with 10 sets randomly selected for training and the remaining 10 sets for testing. The training samples were used to train the classifier to obtain a classification model with optimal parameters. The trained classifier was then used to identify faults in the test samples. The diagnostic results of MPFE, CMPFE, MvMPFE (rms), and CMvMPFE (rms) methods are shown in Figure 10.

The four states are labeled as follows: 1-N, 2-B, 3-IR, and 4-OR. From Figure 10, it can be observed that the MPFE model has three groups of misclassifications: two groups of inner race faults are misclassified as rolling element faults, and one group of outer race fault is misclassified as a rolling element fault. The CMPFE model exhibits two groups of misclassifications, where rolling element faults are mistakenly classified as normal. Similarly, the MvMPFE (rms) model also has two groups of misclassifications, where two groups of normal states are classified as rolling element faults. On the other hand, the CMvMPFE model achieves perfect classification results for all the data, indicating that CMvMPFE (rms) possesses excellent fault representation capability.

Based on Figure 10, it can be inferred that the recognition accuracies of the four models are 92.5%, 95%, 95%, and 100%, respectively. Similarly, CMvMPFE (rms) achieves the highest accuracy among the four models, as shown in Table 1.

Furthermore, considering precision, recall, and F1-Score as evaluation metrics, the confusion matrix for each of the four models, based on their recognition results, is presented in Figure 11.

Since each sample has equal weight, we choose the micro-average metric. From the confusion matrix, we can obtain the precision (P), recall (R), and F1-score (F1) of the four models, as shown in Table 2.

When the confusion matrix is a square matrix, we have P = R = F1. From Table 2, we can conclude that CMvMPFE (rms) achieves the highest score, indicating the best performance or superior fault representation capability.

### 3.5. Comparing Computational Costs

Compared with existing methods, the CMvMPFE method proposed in this paper has the following advantages in terms of computational cost:(1)Computational resource requirements: the method in this paper is simple and effective, with low hardware cost requirements, and the calculation software only MATLAB 2022b required.(2)Model size cost: The CMvMPFE method has a small model and fast calculation speed. The calculation time of 12 groups of feature curves is shown in Figure 12. From Figure 12, it can be seen that the time spent on each calculation is basically consistent, the program is stable, and the calculation is reliable.(3)Algorithm complexity and tuning cost: The algorithm of the method proposed in this paper is simple, and mainly based on the existing fuzzy entropy and permutation entropy, as well as the concept of composite coarse-graining. The subsequent tuning cost is small, and the scale factor, permutation dimension, etc., can be optimized and adjusted. The syntax logic of the algorithm can also be optimized and improved to further speed up the calculation.

## 4. Robustness Analysis of CMvMPFE (rms)

In the fault diagnosis process, it is essential that the fault features are not only capable of effectively representing faults but also possess universality and robustness. This enables the fault extraction feature method to be applied across various scenarios and working conditions, making it an excellent method for fault diagnosis.

### 4.1. Feature Signal Extraction of the Same Fault Type at Different Distribution Positions

For verifying the same outer ring fault, we selected feature signals from different distribution positions and performed composite multi-element multi-scale permutation fuzzy entropy calculation and analysis on the three positions of 3 o’clock, 6 o’clock, and 12 o’clock. The analysis revealed that the composite multiscale permutation fuzzy entropy based on root mean square provided the best results. Therefore, we selected CMvMPFE (rms) as the feature signal extraction method, and the results are presented in Figure 13.

Based on the findings presented in Figure 13, the entropy value distribution of the outer ring faults at the 3 o’clock and 6 o’clock positions show a similar trend, where the values increase first and then decrease with the scale factor. However, the entropy value distribution of the outer ring fault at the 12 o’clock position follows a different pattern, where the values decrease first and then increase and decrease with the scale factor. These results demonstrate that CMvMPFE (rms) is not only capable of distinguishing different state rolling bearing faults, but also effectively identifies faults at different distribution positions, i.e., 3 o’clock, 6 o’clock, and 12 o’clock set on the outer ring, indicating the method’s robustness.

### 4.2. Extraction of Characteristic Signals at the Same Fault Location at Different Fault Depths

Effective fault characteristics should not only be capable of identifying the presence of a fault but also distinguishing between different fault states and their distribution at various locations. Furthermore, it should be able to differentiate between fault depths at the same location. In this study, vibration signal data of the inner ring of a rolling bearing at a speed of 1797 r/min were analyzed under five different working conditions: normal state, and fault depths of 7 mils, 14 mils, 21 mils, and 28 mils. The entropy value of the characteristic parameters of the five different fault depths were calculated using CMvMPFE (rms), and the results are presented in Figure 14.

As illustrated in Figure 14, the distribution of entropy values in the normal state initially increases with the scale factor and then remains stable. On the other hand, the entropy value trend for inner ring fault characteristics at various depths decreases as the scale factor increases. The CMvMPFE (rms) method can effectively differentiate between normal and fault states and even distinguish between different fault depths, resulting in a satisfactory outcome.

## 5. Conclusions

(1)This paper introduces a new entropy algorithm called CMvMPFE (rms) for characterizing the state of vibration signals. The proposed algorithm aims to address the limitations of the incomplete extraction of fault characteristic information under a single scale and the low calculation accuracy and poor anti-interference of multiscale fuzzy and multiscale permutation entropy. The experimental results demonstrate that CMvMPFE (rms) can accurately and completely extract the fault characteristic information of vibration signals. Additionally, the obtained entropy values exhibit better consistency, accuracy, and stability.(2)This paper proposes a new method for diagnosing faults in rolling bearings. The method is based on CMvMPFE (rms) and SVM, and experimental data are used for calculation and comparative analysis. The results demonstrate that the proposed method outperforms existing methods in terms of fault characteristic extraction and pattern recognition accuracy.(3)To verify the robustness and anti-interference ability of the entropy calculation method proposed by CMvMPFE (rms), in this paper, three situations were analyzed and calculated. These included the extraction of characteristic signals at different distribution locations for the same fault type, the extraction of characteristic signals at different fault depths at the same fault location, and the extraction of fault characteristics at different speeds at the same fault depth. Results indicate that the new method proposed in this paper exhibits good robustness and strong anti-interference ability.

However, only three coarse-grained forms is still not comprehensive enough, and the coarse-grained forms need to be expanded.

## Figures and Tables

**Figure 1 entropy-25-01049-f001:**
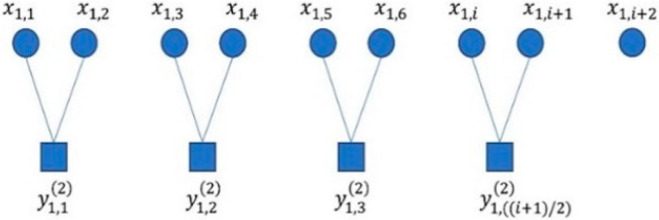
The coarse-graining process of MPFE when τ = 2.

**Figure 2 entropy-25-01049-f002:**
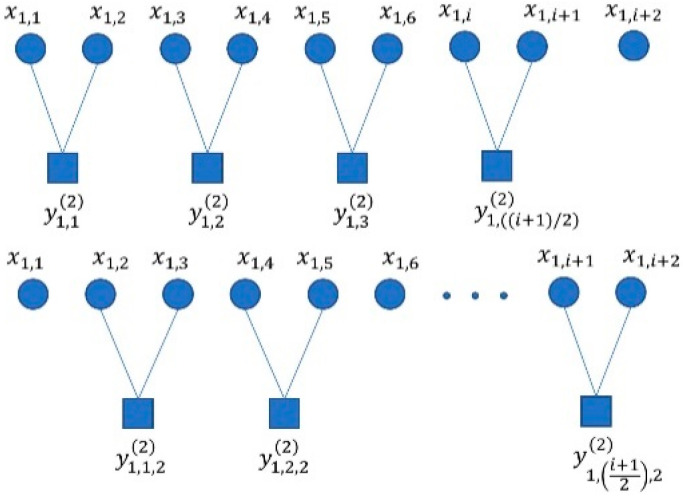
The coarse-graining process of CMvMPFE whe τ = 2.

**Figure 3 entropy-25-01049-f003:**
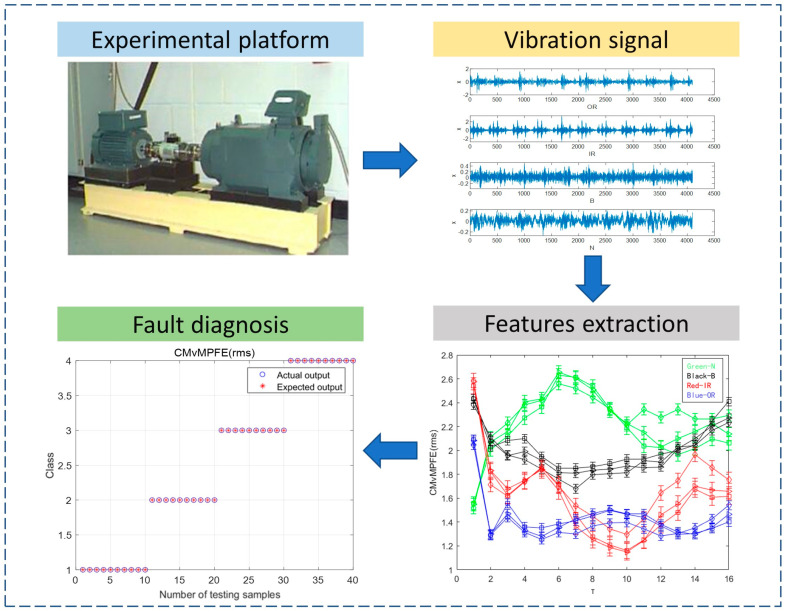
Fault diagnosis process.

**Figure 4 entropy-25-01049-f004:**
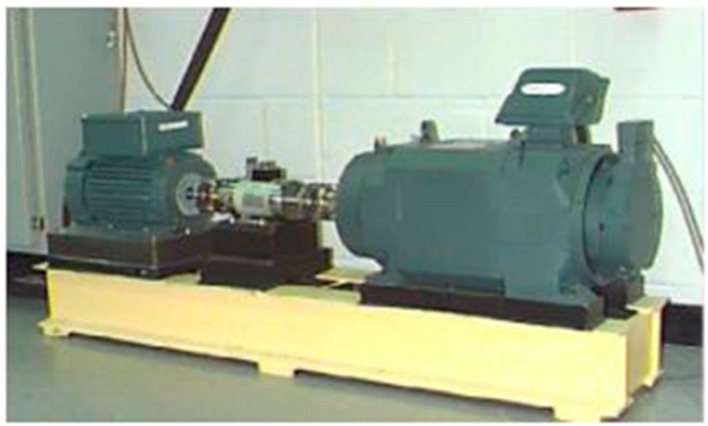
Rolling bearing test system.

**Figure 5 entropy-25-01049-f005:**
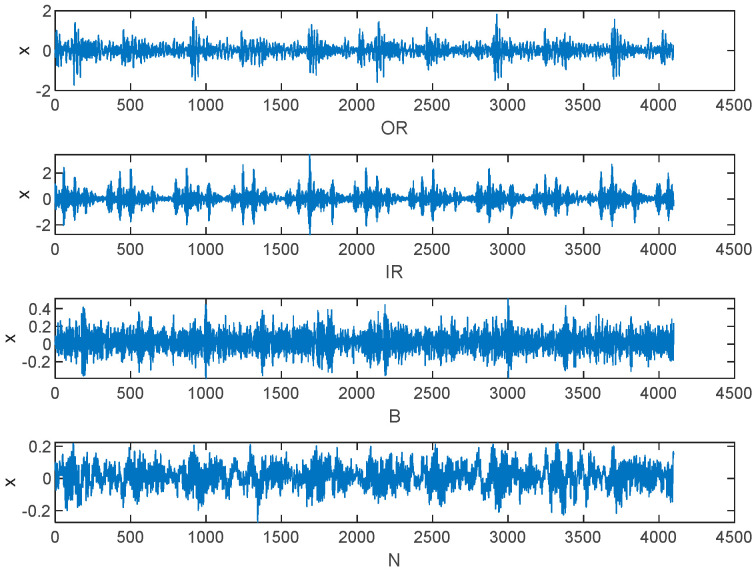
Vibration signals of four states.

**Figure 6 entropy-25-01049-f006:**
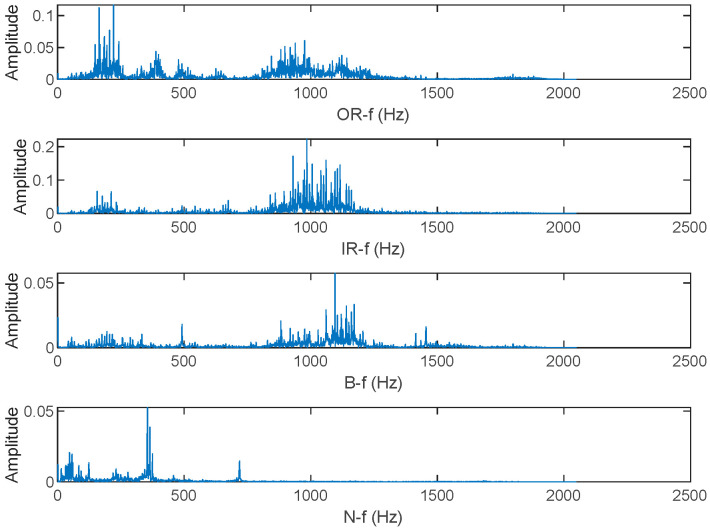
Spectrogram of vibration signals of four states.

**Figure 7 entropy-25-01049-f007:**
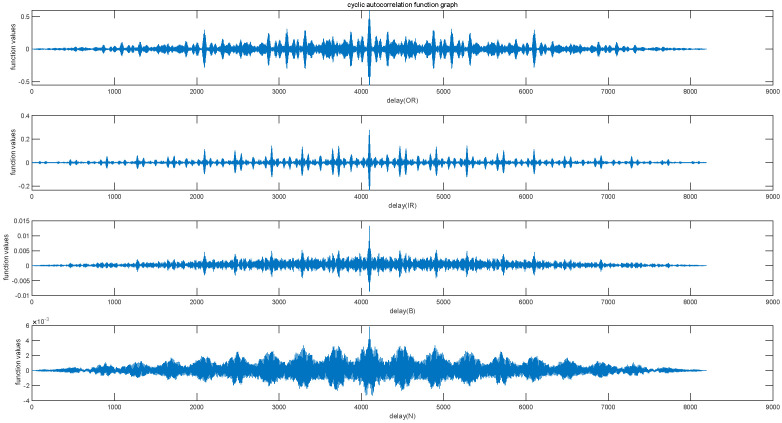
The cyclic autocorrelation function graph of four state signals.

**Figure 8 entropy-25-01049-f008:**
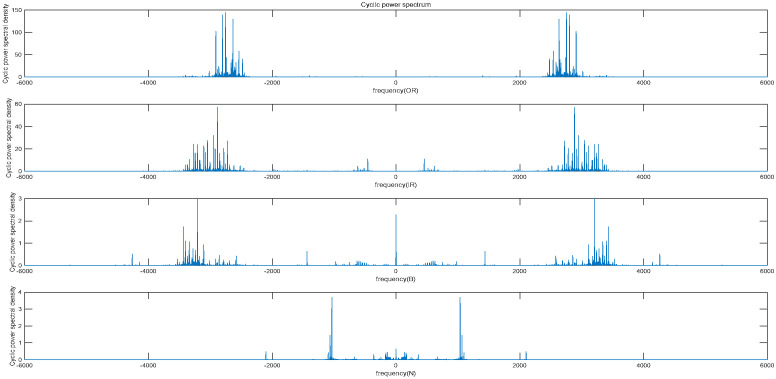
The cyclic power spectrum of four state signals.

**Figure 9 entropy-25-01049-f009:**
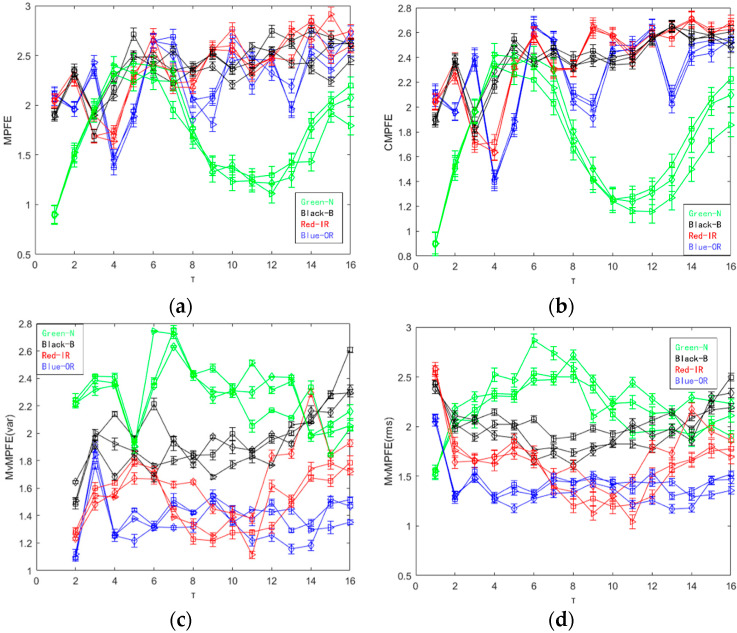
MPFE, CMPFE, MvMPFE (var), MvMPFE (rms), CMvMPFE (var), and CMvMPFE (rms) values of sample data: (**a**) MPFE; (**b**) CMPFE; (**c**) MvMPFE (var); (**d**) MvMPFE (rms); (**e**) CMvMPFE (var); (**f**) CMvMPFE (rms).

**Figure 10 entropy-25-01049-f010:**
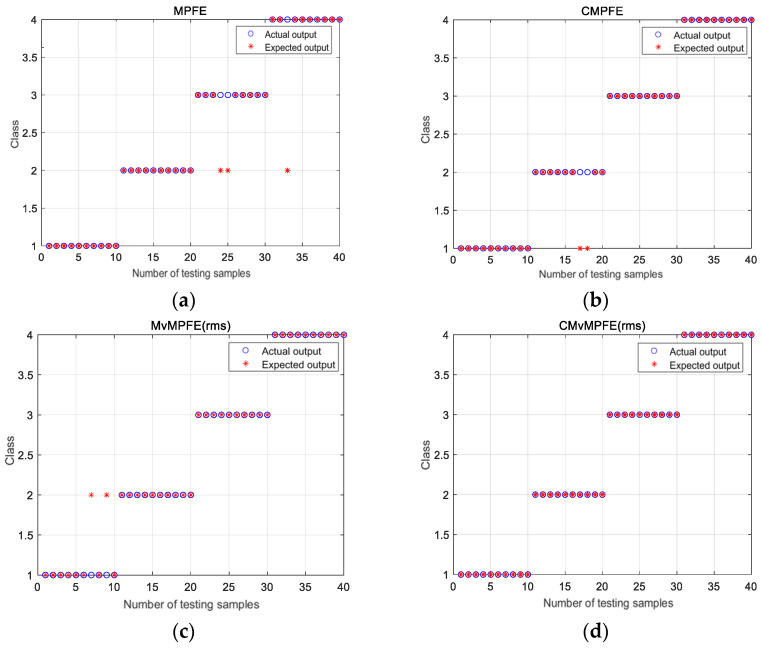
MPFE, CMPFE, MvMPFE (rms), and CMvMPFE (rms) identification results: (**a**) MPFE; (**b**) CMPFE; (**c**) MvMPFE (rms); (**d**) CMvMPFE (rms).

**Figure 11 entropy-25-01049-f011:**
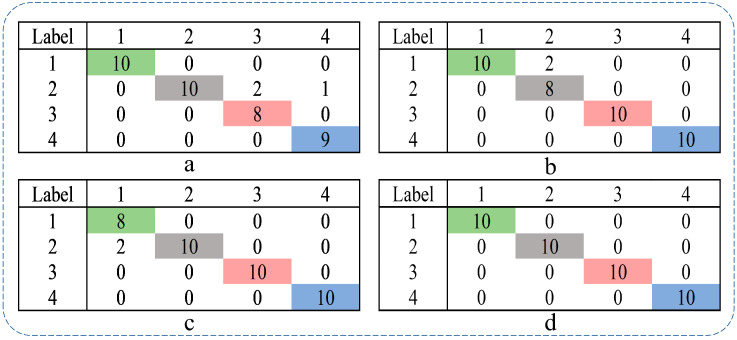
MPFE, CMPFE, MvMPFE (rms), and CMvMPFE (rms) confusion matrix: (**a**) MPFE; (**b**) CMPFE; (**c**) MvMPFE (rms); (**d**) CMvMPFE (rms). (Green indicates the number of normal classified as normal, gray indicates the number of rolling element faults classified as rolling element faults, red indicates the number of inner ring faults classified as inner ring faults, and blue indicates the number of outer ring faults classified as outer ring faults).

**Figure 12 entropy-25-01049-f012:**
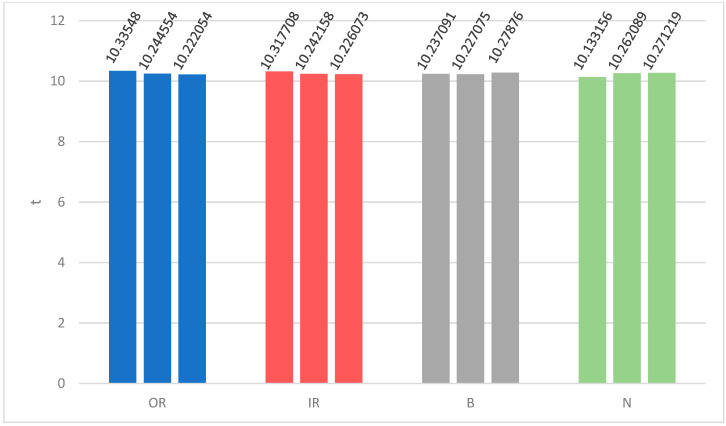
The calculation time of each feature curve for four states.

**Figure 13 entropy-25-01049-f013:**
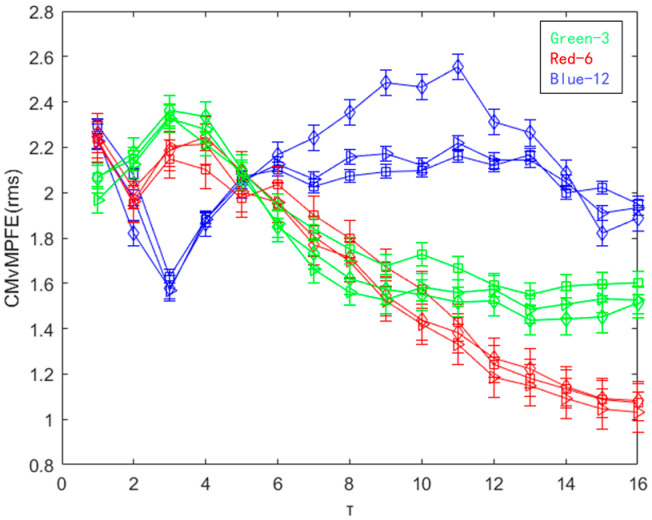
CMvMPFE (rms) of outer ring faults at different distribution positions.

**Figure 14 entropy-25-01049-f014:**
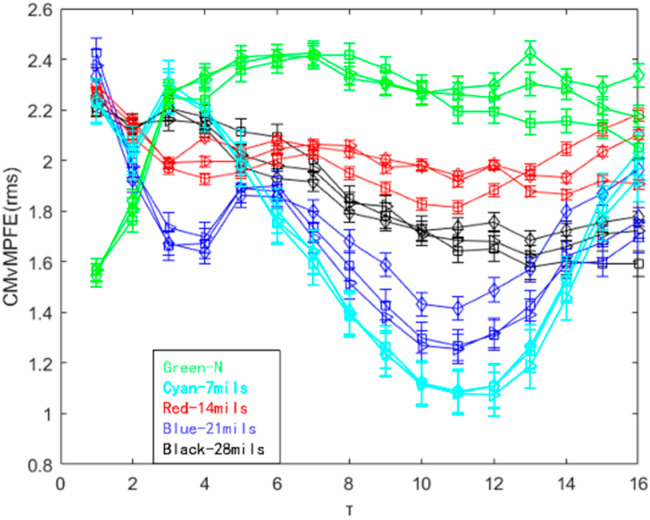
CMvMPFE (rms) of different fault depths of the inner ring at the same speed.

**Table 1 entropy-25-01049-t001:** Identification accuracy.

Feature Models	Accuracy Rate
MPFE	92.5%
CMPFE	95.0%
MvMPFE (rms)	95.0%
CMvMPFE (rms)	**100%**

**Table 2 entropy-25-01049-t002:** P, R, and F1 values for the four models.

Model	P	R	F1
MPFE	0.925	0.925	0.925
CMPFE	0.95	0.95	0.95
MvMPFE (rms)	0.95	0.95	0.95
CMvMPFE (rms)	1	1	1

## Data Availability

The data in this paper come from a publicly available dataset provided by the Bearing Data Center of Case Western Reserve University in the United States. Data links here: https://engineering.case.edu/bearingdatacenter (accessed on 20 September 2022).

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
