# Peer review of "Use of Composite Multivariate Multiscale Permutation Fuzzy Entropy to Diagnose the Faults of Rolling Bearing"

_entropy, 2023, doi:10.3390/e25071049_

Round 1
Reviewer 1 Report
The study focuses on the fault signals of rolling bearings. The Composite Multivariate Multiscale Permutation Fuzzy Entropy (CMvMPFE) method was proposed to solve the problems of low accuracy, large entropy perturbation, and information loss in the calculation process of fault feature parameters. To validate the proposed method, the authors utilized for analyzing experimental data obtained from the Bearing Data Centre of Case Western Reserve University in the United States. The calculation results seems to indicate that the CMvMPFE method can comprehensively and accurately identify bearing states, and has good fault feature extraction capability and robustness.
Remarks:
Please explain the formula from line 128.
Line 238 - a diameter of 21 miles?
The list of references can be improved by adding new works in the field, from an international level.
In the first half of the work, the same ideas are repeated several times.
The results seem to indicate the advantages of the CMvMPFE method. However, not many results are presented with arguments in this regard. Certain numerical indicators that emphasize this aspect should be used.
Reviewer 2 Report
The paper is concerned with a fault detection and isolation approach applied to rolling bearings and based on entropy calculation of vibration signals, which is an interesting topic from both point of views, research and field applications. The paper is well structured being easy for readers to follow the authors’ ideas and understand the theoretical background underlying the proposed approach. A suitable survey of published material related with the paper’s topic is include in the References section and appropriately mentioned in the text. The authors’ proposed approach robustness was evaluated comparing the results obtained with the results achieved using other already reported methodologies. However, in the reviewer opinion the paper to be recommendable for publication needs few changes according to the following comments:
- Throughout the paper there are few typos that should be corrected: references mentioned in text using the upper script format, which should be removed; missed spaces between words and references that should be included; superfluous spaces before commas that should be remove; use of capital letters where lower case should be used (“?(?),?=1,2,…,?, Perform phase”);
- The sentence where Figure 1 is mentioned in the text should be moved to immediately before the mentioned figure;
- The word Reference used in line 201 should be removed;
- Figure 3 should be mentioned in the text;
- The authors claim that “This paper presents a novel method for calculating permutation fuzzy entropy by combining the high calculation accuracy of fuzzy entropy with the strong noise resistance of permutation entropy”. However, never is explained what is meant by “fuzzy entropy”. Thus, this concept should be clarified;
- The authors claim that “… rolling bearing failures are inevitable and can lead to product quality problems or damage and casualties of property.”. However, the mentioned abnormal situations only can be avoided if there is the capability to provide a fault diagnosis in a very early stage of the fault development. Thus, it should be included in the paper a discussion about how the authors’ proposed approach could be used to perform real time fault diagnosis and how it is expected to cope with incipient faults.
Few typos should be corrected as mentioned above.
Reviewer 3 Report
This paper proposed a composite multivariate multiscale permutation fuzzy entropy method for bearing fault diagnosis. Also, the multivariate coarse-grained form was introduced, and the coarse-grained process was improved. Overall, this paper is well structured, and some promising results are presented. Here are some suggestions for further improving the quality of this paper:
1) In the Abstract, lots of technical details are introduced, which might mix the main contribution of this work. Thus, it is suggested to improve the Abstract and highlight the main contributions of this research work.
2) It is suggested to briefly introduce the signal characteristics of rolling bearing, such as impulsiveness and cyclostaionarity, as introduced in the data-driven prognostic scheme for bearing based on a novel health indicator and gated recurrent unit networks.
3) The authors did a good literature review, and the development of entropy-based fault diagnosis has been well described. There are some emerging areas of entropy or its other versions-based reliability analysis that can be included and discussed, especially in terms of novel vibration indicators to monitor gear natural fatigue pitting propagation and novel vibration-based prognostic scheme for gear health management in surface wear progression of the intelligent manufacturing system.
3) At the end of the Introduction, it is suggested to include one paragraph to introduce the whole organization of this paper.
4) Please include a pseudo code to present the developed methodology briefly.
5) It is suggested to include the comparisons of the computation costs with the existing methodologies.
Round 2
Reviewer 1 Report
The paper focuses on the fault signals of rolling bearings. The Composite Multivariate Multiscale Permutation Fuzzy Entropy method was proposed to solve the problems of low accuracy, large entropy perturbation and information loss in the calculation process of fault feature parameters. The results indicate that the method proposed in this paper exhibits good robustness.
The authors took into account the observations from the previous reviews and made additions to the work, both in the introductory part and references, as well as in the analysis of the obtained results.
Reviewer 2 Report
Almost all comments to the paper's previous version were addressed in the preparation of the current version improving its quality and, hence, I recommend the paper for publication as it is.